# Enhanced Microwave Absorption Bandwidth in Graphene-Encapsulated Iron Nanoparticles with Core–Shell Structure

**DOI:** 10.3390/nano10050931

**Published:** 2020-05-12

**Authors:** Danfeng Zhang, Yunfei Deng, Congai Han, Haiping Zhu, Chengjie Yan, Haiyan Zhang

**Affiliations:** 1School of Computer Science and Technology, Guangdong University of Technology, Guangzhou 510006, China; 2School of Materials and Energy, Guangdong University of Technology, Guangzhou 510006, China; dyf1012gdut@163.com (Y.D.); hca0109gdut@163.com (C.H.); HpZhu321@163.com (H.Z.); Richaryan@163.com (C.Y.)

**Keywords:** microwave absorption, graphene-encapsulated iron nanoparticles, permittivity and permeability, simulation, reflection loss

## Abstract

Graphene-encapsulated iron nanoparticles (Fe(G)) hold great promise as microwave absorbers owing to the combined dielectric loss of the graphene shell and the magnetic loss of the ferromagnetic metal core. Transmission electron microscopy (TEM) revealed transition metal nanoparticles encapsulated by graphene layers. The microwave electromagnetic parameters and reflection loss (R) of the Fe(G) were investigated. Graphene provided Fe(G) with a distinctive dielectric behavior via interfacial polarizations taking place at the interface between the iron cores and the graphene shells. The R of Fe(G)/paraffin composites with different Fe(G) contents and coating thickness was simulated according to the transmit-line theory and the measured complex permittivity and permeability. The Fe(G)/paraffin composites showed an excellent microwave absorption with a minimum calculated R of −58 dB at 11 GHz and a 60 wt% Fe(G) loading. The composites showed a wide bandwidth (the bandwidth of less than −10 dB was about 11 GHz). The R of composites with 1–3 mm coating thickness was measured using the Arch method. The absorption position was in line with the calculated results, suggesting that the graphene-coated iron nanoparticles can generate a suitable electromagnetic match and provide an intense microwave absorption. Excellent Fe(G) microwave absorbers can be obtained by selecting optimum layer numbers and Fe(G) loadings in the composites.

## 1. Introduction

With the rapid development of electronic and wireless technologies, electromagnetic interference (EMI) and pollution are becoming serious issues worldwide, affecting more and more electronic integrated devices and the living environment [1,2]. In order to solve these ever-growing serious environment issues, absorbing materials are required to present a number of characteristics (e.g., wide absorption bandwidths, intense absorption, lightweight, and low thickness), the so-called “thin, light, wide, and strong” requirements. Therefore, electromagnetic absorption materials with wide bandwidth, high absorption rate, and tunable electromagnetic properties have been increasingly favored by researchers [3,4]. Among these materials, it is particularly important that the new microwave absorbing materials have appropriate overall performance [5,6]. Ferromagnetic metals and their oxides are well-known for having excellent magnetic loss absorbance [7,8]. However, the chemical instability and heavy mass of ferromagnetic metals in air hinder their potential application as microwave absorbers. Owing to its oxidation resistance, carbon has been proposed as an ideal protective layer material for metals, especially metal nanoparticles [9,10,11,12]. Carbon-based materials also present good dielectric loss along with light weight and broadband microwave absorption. A new nano-structured composite microwave absorption materials such as carbon-encapsulated metal nanoparticles have been found to present improved electromagnetic absorption and tunable electromagnetic properties. This material consisted of a metal nanoparticle inner core and a carbon outer shell. Materials of the carbon shells include graphite [13], amorphous carbon [14], and graphene [15], while metals include Ni [14], Co [16], FeNi [17], FeCo [18], FeSn_2_ [19], and TiO_2_ [20], among others. As we all know, controlling the composition and structure of the composite core and shell can make the dielectric and magnetic properties of these materials more conducive to impedance matching [21]. The dielectric and surface chemistry characteristics of carbon make it a promising shell material for nanoparticles [18,22,23]. Carbon-encapsulated metal nanoparticle structure composite materials present numerous advantages, including wide electromagnetic absorption bandwidth, low density, and stable physical and chemical properties [24]. This outer nanocarbon shell prevents the inner nanometallic particles from oxidation and agglomeration, and the core–shell structure is also conducive to the impedance matching of the enclosed space. As a result, this material showed improved electromagnetic wave absorption properties [25]. Previous reports revealed (Fe, Ni)/C nanocapsules to have improved electromagnetic wave absorption upon proper core composition design and good electromagnetic match [17]. The presence of heterogeneous interfaces and the distinctive core–shell structure increased the number of surface anisotropies and reduced the eddy current in (Fe, Ni)/C as compared with their metal and alloy counterparts. Owing to the unique core–shell structure of the nanoparticles, the electromagnetic field was coupled to the metal magnetic core and the dielectric shell, thus the core–shell structured nanoparticles showed an obviously improved impedance match [26,27].

Core–shell nanostructures have shown excellent electromagnetic properties and microwave absorption characteristics compared with single metal or pure carbon materials [28]. Liu et al. [29] prepared carbon-encapsulated FeNiMo alloy nanoparticles. The excellent electromagnetic properties of this material were the result of the highly complex magnetic permeability and good impedance matching of the FeNiMo core nanoparticles. Zhang et al. [16] discovered multi-dielectric polarizations when studying the wave-absorbing properties of core–shell graphite-coated cobalt nanoparticles. These dielectric polarizations originate from the highly polarized graphite shell and core–shell interface polarization. Liu et al. [30] studied electromagnetic wave absorption properties of FeNi_3_ nanoparticles coated with graphite shells and found that graphite shells increased the magnetic/dielectric loss and attenuation constant of carbon-coated FeNi_3_ nanoparticles and coatings. The core–shell structured carbon-encapsulated metal nanoparticles have shown superior electromagnetic properties compared with traditional absorbers. However, the preparation of thin materials combining high-efficiency microwave absorption and a wide effective bandwidth remains highly challenging, and the electromagnetic loss mechanism of core–shell structured nanoparticles requires further studies. In terms of this view, graphene is a new nanocarbon material that can be used for this purpose. Jian et al. [31] successfully prepared nanoscale Fe_3_O_4_/graphene capsule (GC) composites. The microwave-absorbing characteristics of the as-prepared composites showed a minimum reflection loss (R) of −32 dB at 8.76 GHz and an R lower than −10 dB for an absorption bandwidth of 5.4–17 GHz. The electromagnetic absorption characteristics of graphene-encapsulated magnetic metal nanoparticles have been scarcely studied. Fe(G) materials are largely unexploited despite that they can present a better match between ε and µ than iron absorbers. In this article, we reported simulated and actual measured R values of Fe(G) and compared these values with those previously published (mostly calculated results). As detailed herein, the lower R value of Fe(G) over a wide frequency range can be achieved by optimizing the thickness of the coating material and the concentration of the Fe(G). The electromagnetic characteristics of the Fe(G) were investigated by measuring the electromagnetic parameters of the Fe(G)/paraffin composites. We calculated the R of Fe(G)/paraffin composites with different Fe(G) concentrations and sample thicknesses (1–3 mm). Experimental R measurements were conducted over Fe(G)/epoxy coatings and the results were compared with the calculated R values.

## 2. Experimental

The Fe(G) nanoparticles were prepared by an arc discharge method in our laboratory [32]. The arc discharge was generated by applying a direct current of 150 A at 60 V between two electrodes at an argon pressure of 10 kPa. The distance between the electrodes was 3–4 mm. A graphitic and iron powder of micron size (µm) was used as a raw material. The mixture was shaped in form of cylindrical anode of 25 mm in diameter and 50 mm in height. This anode was consumed and produced soot during the arc discharge process. This soot was deposited on the inner surface of the reaction chamber. After the arc discharge reaction, the soot was collected and Fe(G) nanoparticles samples were obtained.

The microstructure of the Fe(G) nanoparticles was characterized by transmission electron microscopy (TEM, Talos F200S *, Brno, Czech Republic), Raman spectroscopy (Raman, LabRAM HR Evolution type, HORIBA Jobin Yvon, Paris, France), and X-ray diffraction (XRD, D8 Advance type, Germany Bruker Co. Bavaria, Germany). X-ray photoelectron (XPS, Escalab 250Xi type, Thermo Fisher, Massachusetts, America) was used to study the phase composition of the samples and the bond structure of the graphene shells.

Fe(G) and paraffin were mixed in a mechanical mixer at 140 °C to melt the paraffin, and thus achieve homogenous dispersion of Fe(G). The paraffin is an electrical insulator (melting temperature: 92 °C) of nonmagnetic material transparent to electromagnetic waves [33]. The mixture was pressed through a mold into a ring-shaped sample with a thickness of 2–3 mm for the EM wave measurement. The relative permittivity and permeability were obtained by the AV3618 Network Analyzer (Zhongdianke Instrument Co., Ltd. Qingdao, China).

Fe(G) nanoparticles were used as fillers, epoxy resin was used as matrix, and absolute ethanol was used as dispersion medium. Fe(G) nanoparticle slurry was dispersed in epoxy resin to prepare Fe(G) nanoparticle/epoxy resin mixture, and then these coating samples were coated on a 180 mm × 180 mm standard aluminum plate to prepare the measured wave absorption coating. The measured R value was measured by using the arch method, which was one of the important parameters for evaluating the actual reflectance of the absorbing material [34,35]. The aluminum plate operating within 2–18 GHz is considered a “perfect” reflection or 0 dB level (the reference level) material. Therefore, there was no substrate effect on the measured R values.

## 3. Results and Discussion

### 3.1. Characterization of Fe(G)

TEM images of Fe (G) nanoparticles are shown in Figure 1. The Fe (G) nanoparticles were 30–100 nm in diameter, with a wide size distribution (Figure 1a). The Fe(G) nanoparticles showed a spherical morphology and a core–shell structure, with the inner iron core being completely encapsulated by the outer graphene shells. High-resolution TEM (HRTEM, Figure 1b) revealed outer graphene shells of ca. 2–2.5 nm in thickness (ca. 5–7 graphitic layers with a distance of 0.34 nm each layer). Carbon-coated metal nanocapsules with carbon shells of 5–6 nm in thickness have been previously reported [14].

The XRD pattern and the Raman spectrum of the Fe(G) nanoparticles are shown in Figure 2. Fe(G) showed iron and carbon diffraction peaks, and no diffraction peaks of iron oxides or iron carbide were identified. Thus, the Fe core in the Fe(G) nanoparticles remained reduced owing to the protective graphitic shell [17,36]. The TEM image shown in Figure 1 also confirmed the correctness of the XRD results. Three sharp peaks were observed in the XRD pattern corresponding to the (110), (200), and (211) planes of iron, respectively. These data confirmed that the inner iron core of Fe(G) nanoparticles possessed a high degree of crystallinity. The diffraction peaks of the (002) plane of graphene were very weak compared with the peak of Fe, which was not observed in the XRD pattern, and in line with previous works showing no detectable peaks of carbon [17,21,37]. The characterization of carbon species in carbon-based materials is usually identified by Raman spectroscopy. Raman spectra of carbon material usually exhibit two broad peaks at ca. 1350 cm^−1^ (D peak for ‘disordered’ carbon) and at 1580 cm^−1^ (G peak for ‘graphite carbon’). Figure 2b shows the Raman spectrum of the Fe(G) nanoparticles. The bands at 1347 cm^−1^ and 1578 cm^−1^ are characteristic spectra of carbon materials and can correspond to the D and G bands, respectively. This observation is consistent with previous reports [38] showing a defective graphitic layer in the Raman spectra of graphene. What is more, the Raman spectra of graphite powders are shown in Figure 2b. There are two prominent peaks at 1330 cm^−1^ and 1574 cm^−1^, and these correspond to the D and G bands, respectively. As we all know, the value of the I_D_/I_G_ ratio can be used to evaluate the degree of disorder. The I_D_/I_G_ value of graphite powders is 0.4244. What is more, the I_D_/I_G_ value of graphite is 0.6887 in the sample of the Fe(G) nanoparticles and higher than that of graphite powders. The higher ratio means a lower degree of graphitization.

In order to determine the surface composition of the Fe(G), XPS characterization was conducted. Appendix A shows the C1s XPS spectra of the Fe(G) nanoparticles. The fitting curves revealed a C-C binding energy of 284.6 eV, indicative of C1s electron of graphite on the surface [39]. We suggested that the peak at 283.6 eV may correspond to the 1s electrons of graphite at the interface of graphene and Fe in the Fe(G) nanoparticles [40]. These observations were in line with the TEM results.

### 3.2. Complex Permittivity of the Fe(G)/Paraffin Composites

The real (ε’) and imaginary (ε”) parts of the permittivity (ε) of Fe(G)/paraffin composites with 30 wt%, 40 wt%, 50 wt%, and 60 wt% Fe(G) loadings under 2–18 GHz are shown in Figure 3. ε’ and ε” both increased with the Fe(G) loading. ε’ and ε” reached maxima of 10.6 and 3.2, respectively, for the Fe(G)/paraffin composite with a 60 wt% Fe(G) loading. The Fe(G)/paraffin composite with 50 wt% Fe(G) showed ε’ and ε” maxima at 8.4 and 2.5, respectively. ε’ decreased slightly as well as ε” with the frequency, while the opposite trend was found for ε”. The peak of the ε” curve appears at a local minimum, which is caused by polarization [18]. Because of the special core–shell microstructure of Fe(G) nanoparticles, Han et al. [18] also confirms the similarity between the dielectric constant spectra of carbon-coated FeCo materials. These pieces of evidence can provide a reasonable explanation for the dielectric constant curves observed in this paper. What is more, at high frequencies, dipole polarization played a major role, but at low frequencies, weak space charge polarization prevailed [21,27,40]. This is consistent with the previously reported ZnO-coated iron nanoparticles [41] and carbon-coated iron nanocapsules [42]. The ε” curves showed some broad dielectric relaxation peaks at ca. 6.0, 8.5, 11, 13.5, 15, and 17.5 GHz as a result of dielectric relaxation and polarizations. Interfacial polarizations taking place at the interface between iron cores and the graphene shells played a dominant role in determining the dielectric behaviour. As reported by Yan et al. [33], interfacial polarization of carbon-encapsulated FeNi_3_ resulted in strong dielectric losses. As previously reported [16,31,36], a relaxation process can also be generated in inner Ni, Co, and FeNi cores as a result of the enhanced electrical resistivity of the cores encapsulated by outer carbon shells.

### 3.3. Complex Permeability of Fe(G)/Paraffin Composites

The real (µ’) and imaginary (µ”) parts of permeability (µ) for Fe(G)/paraffin composites with 30 wt%, 40 wt%, 50 wt%, and 60 wt% Fe(G) loadings under 2–18 GHz are shown in Figure 4. As shown in Figure 4a, μ’ decreased with the frequency over the entire frequency range. μ’ decreased from 1.30, 1.42, 1.54, and 1.67 to 1.09, 1.06, 1.12, and 1.18 under 2–18 GHz for 30 wt%, 40 wt%, 50 wt%, and 60 wt% Fe(G) loadings, respectively. While μ” exhibited a first increasing and then decreasing trend with an extremum value with increasing frequency, as shown in Figure 4b, μ” showed broad and low maxima at about 0.37, 0.29, 0.20, and 0.14 under 8 GHz for 60 wt%, 50 wt%, 40 wt%, and 30 wt% Fe(G) loadings, respectively. This observation is in good agreement with previous results on carbon-encapsulated FeCo nanoparticles with similar nanoparticle microstructures [18]. In view of the special core–shell structure of Fe(G) nanoparticles, the inner iron core was coated with graphene and separated from each other, so the direct exchange interaction between Fe cores could not be considered, and the dipole interaction was the main factor [30,31]. If there was no graphene shell to isolate the metallic Fe cores from each other, the direct contact of the metallic Fe cores would cause the eddy current to increase sharply while the value of μ’ decreases sharply [17]. In the case of Fe(G) nanoparticles, μ’ decreased and μ” remained at a nearly low value constant with frequency, revealing excellent insulation between the metallic iron cores [28]. Because Fe(G) particles are soft magnets with weak magnetic properties, hysteresis loss and domain wall displacement loss can be ignored. The main loss forms of hysteresis loss are hysteresis loss, eddy current loss, domain wall displacement, and natural resonance. Through the previous analysis, owing to the higher resistivity of Fe(G) nanoparticles and the presence of the graphene shell reducing eddy current losses, it can be inferred that the magnetic loss of Fe(G)/paraffin composites was mainly natural resonance [28,36].

### 3.4. Calculated R Values for Fe(G)/Paraffin Composites

We further investigated the electromagnetic wave absorption properties of the Fe (G). According to the transmission line theory [43], the input impedance of the absorber layers is $Z_{\mathrm{in}}\left(K\right)$, (K = 1, 2, …, N). The input impedance of each layer can thus be calculated by the following formula, and the measured electromagnetic factor data were used to calculate the reflection loss R (or R_L_):(1)$$Z_{\mathrm{in}}\left(K\right)=Z_{c}\left(K\right)\frac{Z_{\mathrm{in}}\left(K-1\right)+Z_{c}\left(K\right)\tan h\left[\gamma \left(K\right)d\left(K\right)\right]}{Z_{c}\left(K\right)+Z_{\mathrm{in}}\left(K-1\right)\tan h\left[\gamma \left(K\right)d\left(K\right)\right]}$$
(2)$$R_{L}\left(\mathrm{dB}\right)=20\log\left|\frac{Z_{\mathrm{in}}-1}{Z_{\mathrm{in}}+1}\right|$$

$Z_{c}\left(K\right)$ and γ(K) can be calculated by Equations (3) and (4), respectively:(3)$$Z_{c}\left(K\right)=\sqrt{\frac{\mu_{0}\mu_{r}\left(K\right)}{\varepsilon_{0}\varepsilon_{r}\left(K\right)}}$$
(4)$$\gamma \left(K\right)=\frac{j2\pi f}{c}\sqrt{\mu_{\left(K\right)}\varepsilon_{\left(K\right)}}=\;j\omega \sqrt{\varepsilon_{0}\mu_{0}\varepsilon_{r}\left(K\right)\mu_{r}\left(K\right)}/c$$
where $Z_{c}\left(K\right)$ and γ(K) are characteristic impedance and propagation constant for each layer, respectively; c is the light speed; ω is the angular frequency; ε_0_ and μ_0_ are the vacuum permittivity and the permeability, respectively; and ε_r_(K) and μ_r_(K) are the relative permittivity and permeability of the K layer absorbing materials, respectively. According to the above formula of R, the microwave absorption performance for Fe(G)/paraffin composites can be tuned by the measured electromagnetic factor data (permittivity and permeability).

Appendix A shows the dependence of the calculated R with the frequency for a Fe (G)/paraffin composite of 3 mm in thickness and with a Fe(G) loading of 40 wt%. The calculated R values of Fe(G) nanoparticles, only graphene (G), and only iron nanoparticles (Fe) are compared in Appendix A. The bandwidths of R < −10 dB were 2.6, 5.6, and 9.1 GHz for Fe, G, and Fe(G), respectively. Thus, Fe(G) showed a wider bandwidth compared with Fe and G owing to suitable complementarities between the dielectric and magnetic losses.

Figure 5 shows the dependence of the calculated R with the frequency for a Fe(G)/paraffin composite of 1–3 mm in thickness with different Fe(G) loadings of 30 wt%, 40 wt%, 50 wt%, and 60 wt%, respectively. The R of the composite of 1 mm in thickness was large. The composite of 2 mm in thickness showed a minimum R value, and this minimum was found to remarkably shift towards low frequencies from 18 to 11 GHz upon increasing the Fe(G) loading from 30 wt% to 60 wt%. The bandwidth of R < −10 dB was larger than 10 GHz for composites of 2 mm in thickness. Especially, a minimum R = −58 dB was obtained at 11 GHz for the composite containing 60 wt% of Fe(G), and a bandwidth of R < −10 dB was obtained from 7 to 18 GHz. Similarly, the R of the composite of 3 mm in thickness shifted towards low frequencies from 13 to 7 GHz upon increasing the Fe(G) loading from 30 wt% to 60 wt%. It is worth noting that the Fe(G)/paraffin composite containing 60 wt% of Fe(G) and 2 and 3 mm in thickness exhibited R < −10 dB over the entire X (8–12 GHz) and Ku (12–18 GHz) bands. Thus, the Fe(G) composites prepared herein were suitable to broaden the bandwidth, and showed improved electromagnetic wave absorption properties.

As a result of the graphene shell outer protection, inner metallic particles with a size smaller than the skin-depth were isolated, enhancing the effective incidence to the electromagnetic wave absorbers [37]. Moreover, compared with some carbon-coated nanoparticle absorbents such as carbon-coated nickel [17,36], Fe(G) exhibited significantly wider bandwidth for electromagnetic absorption. The improved electromagnetic absorption performance of Fe(G) may result from the suitable complementarities between the dielectric and magnetic losses. We hypothesized that the compatibility between the graphene shell and the iron core increased the electromagnetic matching in the Fe(G) nanoparticles.

### 3.5. Measured R of the Fe(G)/Epoxy Coatings

The R of an Fe(G)/epoxy coating with a 60 wt% Fe(G) loading and 1, 2, and 3 mm in thickness was measured and the results are shown in Figure 6. The position of the R peak shifted towards lower frequencies and the minimum R decreased with the thickness of the material. For a 2 mm coating, we obtained a minimum R value of −22 dB at 11 GHz and bandwidth of R < −10 dB was obtained for ca. 9–13 GHz. The positions of the minimum R were in line with the calculated results.

## 4. Conclusions

Fe(G) nanoparticles with a core of magnetic iron nanoparticles and a shell of dielectric graphene were prepared by an arc discharge method. The microstructure, phase, and composition of the Fe(G) were examined by TEM, XRD, Raman, and XPS. The Fe(G) possessed a distinctive core–shell structure, with the inner crystal iron core being completely coated by the outer graphene shell, which consisted of 5–7 nm graphite layers. The major electromagnetic absorption mechanism was dielectric loss. The electromagnetic characteristics of the Fe(G) were investigated under 2–18 GHz, and the results showed that the ε’ decreased with the frequency, while the opposite trend was found for ε”. ε’ and ε” both increased with the Fe(G) loading (30 wt%, 40 wt%, 50 wt%, and 60 wt%) at a fixed frequency. The dielectric loss was the main loss mechanism for the electromagnetic waves of the Fe(G). The calculated R showed a value of −58 dB (bandwidth of R < −10 dB under 7–18 GHz) at 11 GHz for the composite containing 60 wt% Fe(G) and 2 mm in thickness. The results revealed that the electromagnetic absorption properties of the Fe(G) can be adjusted by optimizing both the concentration of Fe(G) and the coating thickness. The peak position of the measured R for an Fe(G)/epoxy coating with 60 wt% Fe(G) showed good agreement with the calculated value. The Fe(G) nanoparticles prepared herein can be used as excellent electromagnetic absorption materials for 2–18 GHz radiations or even higher. The core–shell microstructure of the Fe(G) nanoparticles is of significant importance for establishing good electromagnetic match, dielectric loss, and magnetic loss. Both the theoretical and experimental results of the Fe(G) showed that this core–shell structure is very promising to prepare good electromagnetic absorption materials.

## Figures and Tables

**Figure 1 nanomaterials-10-00931-f001:**
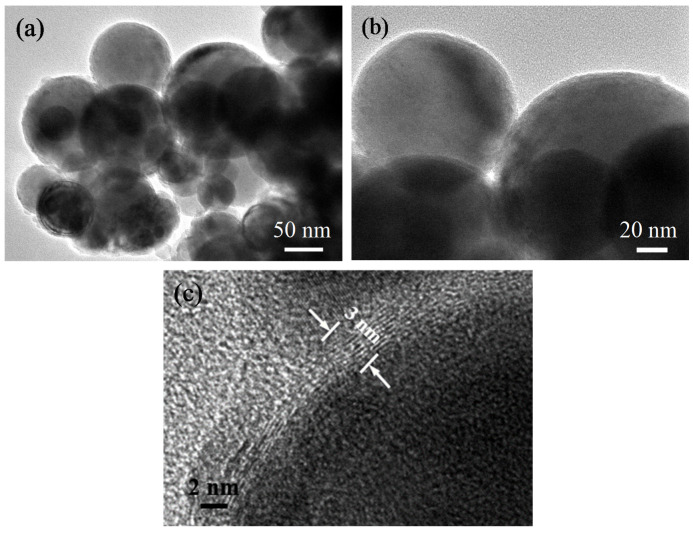
(**a**,**b**) Transmission electron microscopy (TEM) images, (**c**) high resolution TEM (HRTEM) image of Fe(G) nanoparticles.

**Figure 2 nanomaterials-10-00931-f002:**
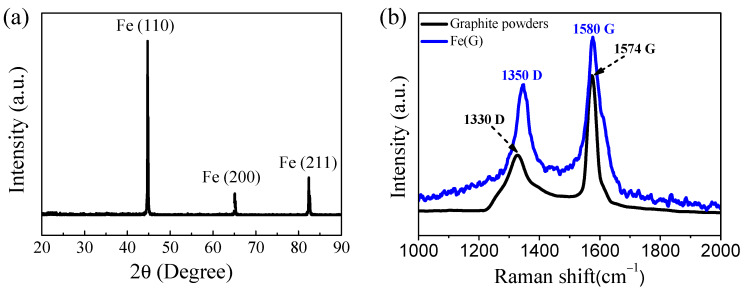
(**a**) X-ray diffraction (XRD) pattern and (**b**) Raman spectrum of Fe(G) nanoparticles and graphite powders.

**Figure 3 nanomaterials-10-00931-f003:**
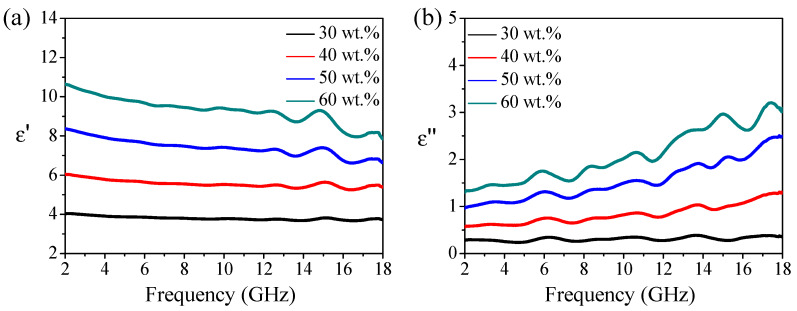
Real (**a**) and imaginary (**b**) parts of complex permittivity of Fe(G)/paraffin composites under 2–18 GHz.

**Figure 4 nanomaterials-10-00931-f004:**
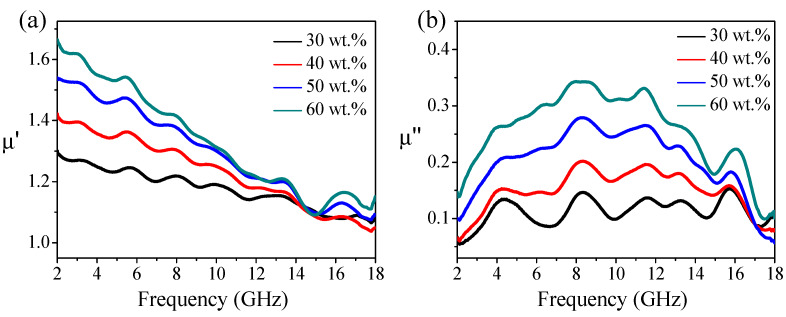
Real (**a**) and the imaginary (**b**) parts of complex permeability for Fe(G)/paraffin composites under 2–18 GHz.

**Figure 5 nanomaterials-10-00931-f005:**
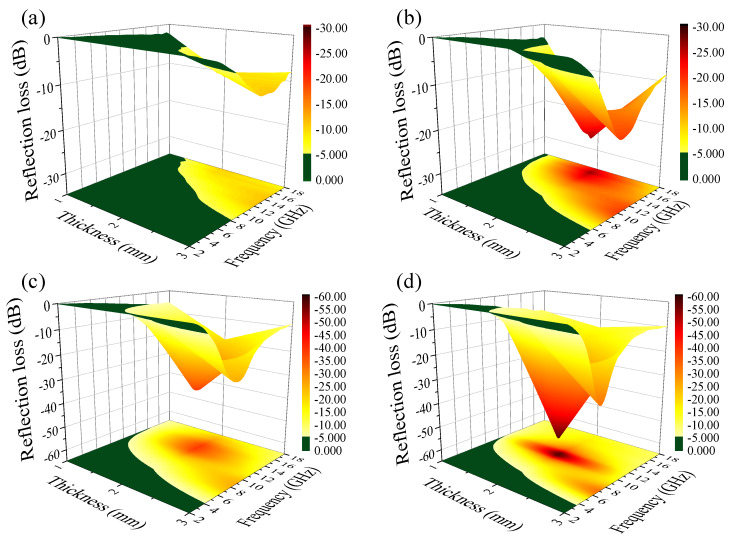
Frequency dependences of R for Fe(G)/paraffin composites with 30 wt% (**a**), 40 wt% (**b**), 50 wt% (**c**), and 60 wt% (**d**) of Fe(G) and 1–3 mm in thickness under 2–18 GHz.

**Figure 6 nanomaterials-10-00931-f006:**
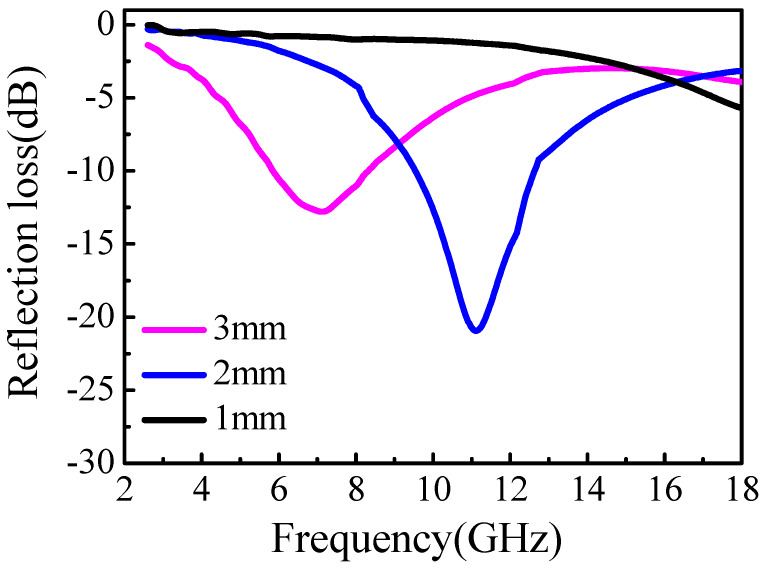
Measured R values for an Fe(G)/epoxy coating with 60 wt% Fe(G) and 1, 2, and 3 mm in thickness.

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
