# Peer review of "Enhanced Microwave Absorption Bandwidth in Graphene-Encapsulated Iron Nanoparticles with Core–Shell Structure"

_nanomaterials, 2020, doi:10.3390/nano10050931_

Round 1

Reviewer 1 Report

The paper describes the microwave absorbing properties of graphene-encapsulated iron nanoparticles. Such particles, combining a metal core and a carbon shell, have shown a great promise in creating materials with enhanced microwave absorption, concerning both the level of reflection losses and the bandwidth of operation in the application-relevant X- and Ku frequency bands. This particular combination, the iron core and the graphene shell, has not been sufficiently studied so far. All these make the content of the paper significant and potentially interesting to the readers. My criticism is mostly of editorial nature (with one exclusion), which is addressed below.

(1) Many typos across the manuscript, e.g. "in lie" (in line) in the abstract, "increasingly number" (2nd line of the introduction, "electromagnetic and electromagnetic absorption" (9th line from below on p. 2), "despite they can present" (8th line from below on p. 2), etc.

(2) Labelling in Figs. 3 and 6 in a too large font.

(3) It must be "real and imaginary parts" not "terms" (11th line from below on p. 5; captions of Figs. 4 and 5).

(4) Incorrect numbering of literature (1st line on p. 6), it should be [40] and [41], not [41] and [42].

(5) Equation (4) is wrong (delete the speed of light and multiply with the imaginary unit). This serious issue must be clarified as equations (1) to (4) form the foundation for the simulation results of the paper.

(6) What is meant by "effective incidence" (2nd line from above on p. 9)?

Author Response

Reviewer 1:Comments and Suggestions for Authors

The paper describes the microwave absorbing properties of graphene-encapsulated iron nanoparticles. Such particles, combining a metal core and a carbon shell, have shown a great promise in creating materials with enhanced microwave absorption, concerning both the level of reflection losses and the bandwidth of operation in the application-relevant X- and Ku frequency bands. This particular combination, the iron core and the graphene shell, has not been sufficiently studied so far. All these make the content of the paper significant and potentially interesting to the readers. My criticism is mostly of editorial nature (with one exclusion), which is addressed below.

Response: Thank you very much for your review. We think highly of the useful comments to improve the quality of our manuscript.

(1) Many typos across the manuscript, e.g. "in lie" (in line) in the abstract, "increasingly number" (2nd line of the introduction, "electromagnetic and electromagnetic absorption" (9th line from below on p. 2), "despite they can present" (8th line from below on p. 2), etc.

Response: Thank you very much for your reminder. We have corrected the typos in the manuscript and marked them in red.

(2) Labelling in Figs. 3 and 6 in a too large font.

Response: Thank you very much for your comments. We have adjusted the size of the Labelling in Figure 3 and Figure 6, and put these two images in the support information according to the comments of the reviewers.

(3) It must be "real and imaginary parts" not "terms" (11th line from below on p. 5; captions of Figs. 4 and 5).

Response: Thank you very much for your suggestions, we have corrected the "terms" in the manuscript to "parts".

(4) Incorrect numbering of literature (1st line on p. 6), it should be [40] and [41], not [41] and [42].

Response: Thanks for your kind suggestion. Thank you for your suggestions. We have corrected the order of the literature in the manuscript.

(5) Equation (4) is wrong (delete the speed of light and multiply with the imaginary unit). This serious issue must be clarified as equations (1) to (4) form the foundation for the simulation results of the paper.

Response: I am sorry, this formula (4) is wrong, it should be multiplied with the imaginary unit "j" and modified to:

= (4)

The formula we input when using software simulation is the correct formula above, otherwise the simulation will not produce the correct result. However, due to the author's mistakes in writing, formula (4) was not multiplied with the imaginary unit "j", and we are sorry again.

(6) What is meant by "effective incidence" (2nd line from above on p. 9)?

Response: Thank you very much for your question. As far as I know, when the electromagnetic wave is directed to the absorber, a part of it will enter the absorber, and at the same time, a part of it will be reflected from the surface of the absorber and cannot enter the absorber. Therefore, we divide the electromagnetic wave into incident electromagnetic wave and reflected electromagnetic wave, and the ratio of the incident electromagnetic wave to the electromagnetic wave is called the "effective incidence".

Reviewer 2 Report

The manuscript reports the synthesis of core/shell iron/graphite nanoparticles. The microwave adsorption properties of the synthesized materials were studied. The issue is of scientific and technological interest and could be of interest to readers of Nanomaterials journal.

I have some points to rise concerning nanomaterials characterization, in order to improve the manuscript and make it publishable.

1) HRTEM analysis: 0.34nm is the typical distance between graphite layer, not the layer thickness.

2) Carbon diffraction peaks are not visible in the XRD spectrum reported in fig.2. Later in the text it is correctly reported that the graphene (002) diffraction peak is hard to detect in XRD spectrum. Please correct.

3) Considering Raman analysis: It could be useful to show also the 2D graphite Raman signal. 

4) Considering XPS analysis it could be useful to clarify the methodology used to fitting C1s spectrum (peak line, background....). Moreover, references supporting the assignment of the peak at 283.6 eV should be reported. Fe 3d spectrum could be very useful to highlight the presence of non-crystalline iron oxides.

Minor point: many typing errors are present, please check carefully whole the manuscript.

Author Response

Reviewer 2:Comments and Suggestions for Authors

The manuscript reports the synthesis of core/shell iron/graphite nanoparticles. The microwave adsorption properties of the synthesized materials were studied. The issue is of scientific and technological interest and could be of interest to readers of Nanomaterials journal.

I have some points to rise concerning nanomaterials characterization, in order to improve the manuscript and make it publishable.

Response: Thank you very much for your evaluation. At the same time, we also think your comments are very helpful to improve the quality of our articles.

1) HRTEM analysis: 0.34nm is the typical distance between graphite layer, not the layer thickness.

Response: Thank you very much for your comments. I apologize for not explaining it clearly. I have changed to “ca. 5–7 graphitic layers with a distance of 0.34 nm each layer” in the manuscript, and marked with red font.

2) Carbon diffraction peaks are not visible in the XRD spectrum reported in fig.2. Later in the text it is correctly reported that the graphene (002) diffraction peak is hard to detect in XRD spectrum. Please correct.

Response: Thank you very much for your question. We want to express that the peak of C is too weak compared to the peak of Fe, so the peak of C cannot be seen on the XRD pattern, which is consistent with the report in [17,21,37]. We have revised it with red font in the article.

3) Considering Raman analysis: It could be useful to show also the 2D graphite Raman signal.

Response: The Raman spectra of graphite powders are added in Figure 2b. There are two prominent peaks at 1330 cm−1 and 1574 cm−1, and these respectively correspond to the D and G bands. As we all known, the value of the ID/IG ratio can be used to evaluate the degree of disorder. The ID/IG value of graphite powders is 0.4244. What’s more, the ID/IG value of graphite is 0.6887 in the sample of the Fe(G) nanoparticles and higher than that of graphite powders. The higher ratio means a lower degree of graphitization.

(b) Raman spectrum of Fe(G) nanoparticles and Graphite powders.

4) Considering XPS analysis it could be useful to clarify the methodology used to fitting C1s spectrum (peak line, background....). Moreover, references supporting the assignment of the peak at 283.6 eV should be reported. Fe 3d spectrum could be very useful to highlight the presence of non-crystalline iron oxides.

Response: Thanks for your kind suggestion. We refer to the literature [39] and [43] to determine the approximate position of the XPS peak, and then use the Xpeak software to fit the peak position. In addition, we have added ref. [43] in the revised manuscript to support the assignment of the peak at 283.6 eV. In addition, due to the Novel coronavirus pneumonia, the Fe 3d spectrum cannot be tested this month. We are very sorry for this.

[43] Feng, J.; Zong, Y.; Sun, Y.; Zhang, Y.; Yang, X.; Long, G.; Wang, Y.; Li, X.; Zheng, X. Optimization of porous FeNi3/N-GN composites with superior microwave absorption performance. Chem. Eng. J. 2018, 345, 441–451.

Minor point: many typing errors are present, please check carefully whole the manuscript.

Response: Thank you very much for your reminder, we have checked our manuscript and corrected the error in red font.

Reviewer 3 Report

The manuscript is enough interesting bit so long for the goals declared in the Abstract. The number of illustrations is very excessive and should be reduced down to 6. XPS data and Raman data are not essential for this research and should be described in words in this research. Just specify the positions and widths of D and G bands and their intensity ratio ID/IG. XPS and Raman spectrum figures are not necessary in the main body of manuscript. Please delete them or shift into the Suppl. Data file. Fig.6 and Fig.7 are also not necessary and can be deleted. Such  excessive data only overcomplicates the reading and understanding. Authors should concentrate only on elucidation of appearance of Fig.8 in their manuscript. The overall content should be made more shortened (on 25%). The revised version of manuscript with no more than 6 figures can be published after the next compulsory reviewing.

Author Response

Reviewer 3:Comments and Suggestions for Authors

The manuscript is enough interesting bit so long for the goals declared in the Abstract. The number of illustrations is very excessive and should be reduced down to 6. XPS data and Raman data are not essential for this research and should be described in words in this research. Just specify the positions and widths of D and G bands and their intensity ratio ID/IG. XPS and Raman spectrum figures are not necessary in the main body of manuscript. Please delete them or shift into the Suppl. Data file. Fig.6 and Fig.7 are also not necessary and can be deleted. Such excessive data only overcomplicates the reading and understanding. Authors should concentrate only on elucidation of appearance of Fig.8 in their manuscript. The overall content should be made more shortened (on 25%). The revised version of manuscript with no more than 6 figures can be published after the next compulsory reviewing.

Response: Thank you very much for your comments. We have reduced the number of illustrations in the article to six, and at the same time reduced the XPS pattern (original figure 3) and The comparison chart (original picture 6) of the R value of graphene (G), iron nanoparticles (Fe) and Fe (G)/paraffin composites is put into the supporting information file. In addition, the position of the specified D and G bands and their intensity ratio ID/IG are also added to the original text. But I think Figure 7 is necessary in this article, because this is one of the important indicators to judge the absorption capacity of the absorbent material, so I kept it in the article.

Round 2

Reviewer 1 Report

The manuscript has been adequatelly corrected.

Reviewer 2 Report

Authors have improved their manuscript following my suggestion